# The role of mycorrhizal type and plant dominance in regulating nitrogen cycling in Oroarctic soils

Aurora Patchett<sup>1,2</sup>, Louise Rütting<sup>3</sup>, Tobias Rütting<sup>1</sup>, Samuel Bodé<sup>4</sup>, Sara Hallin<sup>5</sup>, Jaanis Juhanson<sup>5</sup>, C. Florian Stange<sup>6</sup>, Mats P. Björkman<sup>7,2</sup>, Pascal Boeckx<sup>4</sup>, Gunhild Rosqvist<sup>8</sup>, and Robert G. Björk<sup>1,2\*</sup>

<sup>1</sup>Department of Earth Sciences, University of Gothenburg, Box 460, SE-40530 Gothenburg, Sweden

<sup>2</sup>Gothenburg Global Biodiversity Centre, P.O. Box 461, SE-405 30 Gothenburg, Sweden

<sup>6</sup>Federal Institute for Geosciences and Natural Resources (BGR), Hanover, Germany

Abstract. Mycorrhizal fungi enhance plant access to nitrogen (N) in nutrient-poor environments like the Arctic tundra by depolymerizing N-rich organic compounds into forms available to plants and microbes. As climate change reshapes plant communities and mycorrhizal associations, shifting dominance from herbaceous species to shrubs, changes in mycorrhizal type and plant species dominance may differentially stimulate N cycling. Both dominant and rare species, along with mycorrhizal associations, contribute to ecosystem processes and stability, though the specific roles of these components in N cycling and overall ecosystem functioning remain uncertain. We investigated how mycorrhizal associations and plant diversity affect gross N mineralization and nitrification rates in an Oroarctic ecosystem. Four years after a plant removal treatment, we measured these rates using in situ 15N labelling and quantified a selection of nitrification genes. Treatment plots included (1) unmanipulated (Control); or the removal of: (2) ectomycorrhizal (EcM) and ericoid mycorrhizal (ErM) plants, letting arbuscular mycorrhizal (AM) and non-mycorrhizal (NM) plants dominate (AM/NM); (3) AM and NM plants, letting EcM and ErM plants dominate (EcM/ErM); (4) low-abundance species, leaving the most abundant species (Dominant); and (5) highabundance species, leaving only the low-abundance species (Rare). Gross N mineralization rates were 73 % and 78 % higher in EcM/ErM and Dominant, respectively, compared to Control, while AM/NM and Rare showed more moderate increases of 30 % and 46 %. Gross nitrification was also highest in EcM/ErM, with a 26 % increase over Control. Gene abundances did not mirror nitrification patterns. Archaeal ammonia oxidizers (AOA), Nitrospira-type nitrite oxidizers (NIS), and comammox clade A (ComaA) were consistently more abundant than bacterial ammonia oxidizers (AOB), Nitrobacter-type nitrite oxidizers (NIB), and comammox clade B (ComaB), suggesting a stable site-level nitrifier community. Dominant had the lowest gene copy numbers overall, except for AOB, which was highest. In addition, AOA gene abundance was significantly lower in Dominant compared to Control, with a marginal reduction observed for NIS. Our findings highlight the key role of EcM/ErM fungi in accelerating N cycling in Oroarctic soils, challenging traditional assumptions that N transformation rates are slow in

<sup>&</sup>lt;sup>3</sup>Chair of Soil and Plant Systems, Brandenburg University of Technology Cottbus-Senftenberg, Cottbus, Germany

<sup>&</sup>lt;sup>4</sup>Department of Green Chemistry and Technology, Isotope Bioscience Laboratory, Ghent University, Gent, Belgium

<sup>&</sup>lt;sup>5</sup>Department of Forest Mycology and Plant Pathology, Swedish University of Agricultural Sciences, Uppsala, Sweden

<sup>&</sup>lt;sup>7</sup>Department of Biology and Environmental Sciences, University of Gothenburg, Box 463, SE-40530 Gothenburg, Sweden

<sup>&</sup>lt;sup>8</sup>Department of Physical Geography, Stockholm University, Stockholm, 106 91, Sweden

<sup>\*</sup>Correspondence to: Robert G. Björk (robert.bjork@gu.se)

EcM/ErM dominated ecosystems. These insights underscore the need to consider mycorrhizal associations and plant community composition when predicting tundra ecosystem responses to environmental change.

#### 1 Introduction

35

55

65

The availability of soil nutrients plays a pivotal role in shaping tundra plant productivity and the composition of plant communities (Chapin et al., 1995; Shaver et al., 2001), as well as their responses to climate change (Aerts, 2009; Riley et al., 2021; Stow et al., 2004; Sturm et al., 2001). As climate change is particularly pronounced in the Arctic, shifts in plant growth and community composition are occurring (Bjorkman et al., 2020; Hollister et al., 2015), including increased plant productivity ("arctic greening") and shrub expansion ("shrubification") (Bjorkman et al., 2019; Mekonnen et al., 2021; Myers-Smith et al., 2011; Sistla et al., 2013; Tape et al., 2006). Changes in plant community composition contribute to shifts in biodiversity aboveand below-ground (Mod and Luoto, 2016; Parker et al., 2018, 2021), with mycorrhizal fungi mediating these changes by influencing soil microbial community composition and activity, impacting soil carbon (C) content, and nutrient cycling (Andresen et al., 2022; Bahram et al., 2020; Eagar et al., 2022; Hawkins et al., 2023; Hobbie and Högberg, 2012; Hobbie and Hobbie, 2006; Netherway et al., 2021; Phillips et al., 2013; Read, 1991; Sun et al., 2023; Tedersoo et al., 2020). Thus, changes in plant identity or functional diversity can alter nitrogen (N) availability through indirect effects on N mineralization, nitrification, and other N transformations (Isobe et al., 2018; Robertson and Groffman, 2015). These alterations can feed back to plant growth and enhance ecosystem C cycling (Hicks et al., 2020a, 2022; Mekonnen et al., 2021; Parker et al., 2021). Therefore, understanding the links between plant community composition, soil microorganisms, and N cycling is vital for predicting climate change impacts on tundra ecosystems, yet these interactions remain poorly understood (Dobbert et al., 2022).

Ecological communities are typically composed of a few abundant species and many rarer ones (Gaston, 2011; McGill et al., 2007). Traditionally, research has focused on the role of dominant species in ecosystem functioning, but both dominant and rare species contribute to ecosystem stability and processes (Avolio et al., 2019; Jain et al., 2014; Lyons et al., 2005; Lyons and Schwartz, 2001; Richardson et al., 2012; Säterberg et al., 2019; Smith and Knapp, 2003). According to the mass ratio hypothesis (Grime, 1998), ecosystem processes such as primary production, nutrient cycling, and soil microbial composition are primarily driven by dominant plant species, whose high biomass and resource use exert a disproportionate influence—while the contributions of rare species are considered minimal (Grime, 1998; Tedersoo et al., 2020). This disproportionate influence also extends to N dynamics, where dominant species, through their biomass-scaled traits, can affect soil N availability by regulating N mineralization and nitrification (Clemmensen et al., 2021; Kielland, 1995; Liu et al., 2018; Michelsen et al., 1996; Ramm et al., 2022; Rozmoš et al., 2022; Tunlid et al., 2022). Rare species, in contrast, often exhibit higher functional diversity and may fill ecological roles not occupied by dominant species, facilitating niche differentiation and promoting ecosystem resilience (Dee et al., 2019; Hooper et al., 2005; Leuzinger and Rewald, 2021; Mouillot et al., 2013; Soliveres et

al., 2016; Tang et al., 2023). While their overall biomass contribution is lower, their diverse traits and microbial interactions could play an important role in nutrient partitioning. Both dominant and rare species can form mycorrhizal associations, but differences in mycorrhizal types and plant-microbe interactions may drive variation in N cycling at the community level (Knops et al., 2002; Van der Krift and Berendse, 2001; Moreau et al., 2015, 2019).

70

85

Ectomycorrhizal (EcM) and ericoid mycorrhizal (ErM) fungi tend to dominate Arctic ecosystems (Michelsen et al., 1998; Soudzilovskaja et al., 2017; Steidinger et al., 2019), whereas arbuscular mycorrhizal (AM) fungi are considered less common due to low cold tolerance (Kilpeläinen et al., 2016; Kytöviita, 2005; Ruotsalainen and Kytöviita, 2004; Wang et al., 2002). These three mycorrhizal types differ in their influence on N mineralization rates and inorganic N availability (Björk et al., 2007; Phillips et al., 2013; Tedersoo et al., 2020). AM fungi facilitate rapid N turnover by promoting inorganic N uptake (Govindarajulu et al., 2005; Hodge and Storer, 2015; Savolainen and Kytöviita, 2022), EcM fungi access both organic and inorganic N, leading to intermediate N turnover rates (Hobbie et al., 2009; Kohler et al., 2015; Miyauchi et al., 2020; Orwin et al., 2011; Pellitier and Zak, 2018), and ErM fungi specialize in mobilizing N from complex organic compounds, contributing to slower N cycling (Bending and Read, 1996; Clemmensen et al., 2021, 2024; Fanin et al., 2022; Tybirk et al., 2000; Wurzburger and Hendrick, 2009). In ecosystems dominated by a single mycorrhizal type, nutrient cycling may become increasingly constrained by that symbiosis, leading to homogenized soil N dynamics. For example, EcM fungi effectively access organic-N, stabilizing it in less labile forms and reducing N losses, whereas AM fungi promote greater N mobility. potentially increasing N loss (Hobbie and Ouimette, 2009). In contrast, communities composed of less abundant, locally rare species may support different or complementary N cycling functions compared to those dominated by the most abundant species, potentially enhancing functional redundancy and buffering against environmental fluctuations — even when species richness is held constant. To understand these dynamics, it is essential to disentangle the effects of plant dominance, species diversity, and mycorrhizal associations on N cycling.

We aimed to determine the relative effects of functional (mycorrhizal) and structural (Rare, Dominant) diversity on soil N cycling. To address this, we conducted a plant removal experiment and in-situ <sup>15</sup>N labelling to determine gross N mineralization and nitrification rates, key processes regulating N supply and loss. We also used quantitative PCR (qPCR) to quantify six microbial genes related to nitrification, assessing the genetic potential for this process. Our mycorrhizal groupings reflect broad nutrient cycling patterns (Averill et al., 2014; Read and Moreno, 2003) and finer-scale dynamics (Giesler et al., 1998; Björk et al., 2007). EcM and ErM were grouped based on shared traits such as saprotrophic capacity and organic nutrient acquisition, while AM and non-mycorrhizal (NM) associated plants were linked by their association with faster nutrient turnover. These mycorrhizal types also naturally co-occur in tundra vegetation. To separate the effects of mycorrhizal function from species richness, as argued in the mass ratio hypothesis, we also varied species dominance and rarity, enabling us to test how functional traits and community structure influence ecosystem processes. We hypothesize that (1) gross N mineralization rates will be highest in EcM/ErM-dominated plots, due to their efficiency in accessing organic N sources; (2) gross nitrification rates will

be highest in AM/NM-dominated plots, which promote rapid N turnover and stimulate nitrifier activity; (3) higher gross nitrification rates will correspond with greater genetic potential for ammonia and nitrite oxidation in AM/NM-dominated plots; and (4) mycorrhizal type will exert a stronger influence on N processes than plant community structure, given its direct role in N acquisition and cycling.

#### 2 Methods

#### 2.1 Study site and design

This study was conducted at the Tarfala Research Station in the Tarfala valley of the Kebnekaise Mountains, northern Sweden, at elevations ranging from 1098 to 1114 m a.s.l. (Latitudes:  $67^{\circ}54'14.16''N$  to  $67^{\circ}54'15.16''N$ , Longitudes:  $18^{\circ}37'23.80''E$  to  $18^{\circ}37'29.39''E$ ). The geomorphology of the valley reflects its glacial history, with landforms shaped by retreating ice masses and a substrate dominated by rocky debris. The study area is situated near the terminal moraines marking the maximum extent of Storglaciären during the Little Ice Age (~1910) (Holmlund, 1987) and is characterized by shallow soils developed on till, classified as Leptosols and Regosols (Fuchs et al., 2015). Prominent plant species are the graminoids *Carex bigelowii*, *Carex nigra*, *Deschampsia flexuosa*, *Festuca vivipara*, and *Juncus trifidus*; the deciduous dwarf shrubs *Salix polaris* and *Vaccinium uliginosum*; the evergreen dwarf shrubs *Dryas octopetala* and *Empetrum nigrum*; and the forbs *Bistorta vivipara* and *Silene acaulis*. The mean annual air temperature from 1995 to 2019 was -2.6  $\pm$  1.8 °C with the coldest month in February (-10.5  $\pm$  5.5 °C) and the warmest month in July (8.4  $\pm$  3.7 °C) (SMHI 1995-2019; *raw data retrieved from www.smhi.se*). The summer mean precipitation is  $458 \pm 201$  mm (Dahlke et al., 2012, Tarfala Research Station 1980-2011; *available at https://bolin.su.se/data/tarfala/climate.php*).

We established a plant removal experiment in 2016 with one unmanipulated control and four treatments designed to manipulate plant community structure: 1) Control, where no plant species were removed; 2) AM/NM, where all plants with EcM or ErM associations were removed, leaving only plants with AM or NM; 3) EcM/ErM, where all plants with AM or NM associations were removed, leaving only plants with EcM or ErM associations; 4) Dominant, where rare plant species were removed, leaving the eight most dominant species (9 rare species removed; Table S1, S2); and 5) Rare, where dominant plant species were removed, retaining the eight rarest species (7–11 dominant species removed; Table S1, S2). The Dominant and Rare species removal treatments were designed to include a relatively even mixture of species representing different mycorrhizal types (EcM, ErM, AM, and NM). This design allowed us to separate the effects of species richness from those of mycorrhizal association. While the exact number of species removed varied slightly between treatments, both included a balanced representation of mycorrhizal types. Species removal was performed by clipping vegetation at the soil surface, with treatments maintained from 2016 to 2019 by removing regrowth of undesired species each growing season. The treatments were distributed across 32 plots arranged into eight blocks, each containing four plots (one for each treatment group, except for Rare and Dominant, which were represented in four blocks each). There were eight replicates for the AM/NM-dominant, EcM/ErM-

dominant, and control treatments, and four replicates for the Rare and Dominant plant community treatments. Each consisted of a smaller survey area (1 m²) to exclude edge effect of the trenching (4 m²) designed to exclude external mycorrhizal colonization. Trenches were dug around the 4 m² perimeter and lined with 1 µm mesh to a depth of 0.3 m, allowing water movement but preventing root and mycorrhizal penetration.

#### 2.2 Plot-level plant diversity

160

To determine plot-level plant community structure, we conducted two vegetation surveys (pre-clipping, July 2015 and post-clipping, July 2019) using point intercept measurements (Molau and Mølgaard, 1996)-on the central 1 m<sup>2</sup> quadrats for each plot. In addition, all species within the 4 m<sup>2</sup> plot not registered by point intercept were noted. We estimated the cover and counts of each species to determine species richness. We also calculated the transient changes in community dynamics initiated by altered plant interactions and estimated changes in above-ground biomass (Molau, 2010).

#### 2.3 Nitrogen dynamics

Four years after plant removal, gross soil N dynamics were investigated in the field using the virtual soil core <sup>15</sup>N tracing approach (Rütting et al., 2011) and a mirror <sup>15</sup>N labelling approach, allowing investigation of N transformations in the intact mycorrhizosphere. Within each plot, we set up two groups of four injection locations in opposing corners: one corner for <sup>15</sup>Nlabelled ammonium (NH<sub>4</sub><sup>+</sup>) and the other for <sup>15</sup>N-labelled nitrate (NO<sub>3</sub><sup>-</sup>), to avoid cross-contamination. We conducted the <sup>15</sup>N labelling by injecting <sup>15</sup>(NH<sub>4</sub>)<sub>2</sub>SO<sub>4</sub> (Cambridge Isotope Laboratories) to quantify gross N mineralization or K<sup>15</sup>NO<sub>3</sub> 150 (Cambridge Isotope Laboratories) to quantify nitrification, both labels with a <sup>15</sup>N fraction of 99 %, to a soil depth of 6 cm, both treatments also receiving the unlabelled other moiety. There were 11 injection points per location using a template for guidance (Rütting et al., 2011), each point receiving 1.14 mL of solution containing 15.0 µg NH<sub>4</sub>-N mL<sup>-1</sup> and 4.5 µg NO<sub>3</sub>-N mL<sup>-1</sup>, which is equivalent to c.a. 3.8 µg NH<sub>4</sub>-N g<sup>-1</sup> dry soil and 1.2 µg NO<sub>3</sub>-N g<sup>-1</sup> dry soil. These amounts were calculated based on soil concentrations measured in tundra soil in the nearby Latnjajaure Field Station (Björk et al., 2007), aiming for a <sup>15</sup>N 155 enrichment of 10%. For NO<sub>3</sub>, a larger amount was added, approximately 50% of the native pool, which was required for <sup>15</sup>N analysis. We destructively harvested soil cores at 2, 25, 49, and 97 hours after labelling, using sharpened PVC tubes (3 cm in diameter), inserted to a depth of 6 cm within the organic soil layer, at each of the four respective locations.

Soil cores were immediately processed at the Tarfala Research Station to extract inorganic N (i.e., NO<sub>3</sub><sup>-</sup> and NH<sub>4</sub><sup>+</sup>) following initial sieving (mesh size 2 mm). 10 g of field moist soil were extracted using 20 ml of 1 M KCl and placed on a shaker for 60 min at 250 rpm before filtration with Whatman 1 G/F filter paper (11 µm). The extracts were stored frozen at -20°C until further analyses. Concentrations and the <sup>15</sup>N fraction of NH<sub>4</sub><sup>+</sup> were determined from soil KCl extracts using the micro diffusion technique (Biasi et al., 2022; Brooks et al., 1989), followed by <sup>15</sup>N analysis on an elemental analyzer (Europa EA-GSL, Sercon

Ltd., UK) coupled to an Isotope Ratio Mass Spectrometer (Sercon 20-22, Sercon Ltd., UK). NO<sub>3</sub><sup>-</sup> concentrations and the <sup>15</sup>N fraction in all samples were determined from soil KCl extracts using the SPINMAS technique (Stange et al., 2007). The TN, TC, C:N ratio, and bulk <sup>15</sup>N in soil were measured using the EA-IRMS described above. Dried soil was first ground (Retsch MM400, frequency 23.0 1/s, for 2 min) and around 15 mg from each sample was placed into a tin capsule.

#### 2.4 Soil characteristics

At the same time as the <sup>15</sup>N labelling experiment, we also collected samples from the top 6 cm of the organic soil layer to assess abiotic and biotic soil characteristics, matching the depth used for labelling. Four soil samples (10×10×6 cm, 250 cm<sup>3</sup> each) were collected from each plot after the <sup>15</sup>N labelling experiment to avoid destructive sampling within the plot during the experiment. Stones, plant shoots and roots were removed from the collected samples immediately after sampling at the Tarfala Research Station, which were then sieved through a 2 mm mesh. The four sieved soil samples from each plot were combined and homogenized. Subsamples were separated from the homogenized soil for various analyses elsewhere, including pH, gravimetric soil water content (GWC, g/g), soil organic matter (SOM), and DNA extraction for abundance of microbial communities. Subsamples for DNA extraction were stored frozen until further analyses.

GWC was measured by oven-drying 10 g of wet soil at 100 °C for 24 hours. SOM content was determined using the loss-onignition method by heating the soil at 550 °C for 6 hours. Soil pH was measured in water (10 g soil, 1:1 deionized water) and in 1 M KCl (10 g soil, 1:4). Field measurements of soil temperature at a depth of - 5 cm ( $T_{\text{soil}}$  °C) and soil water content at 0 – 6 cm (volumetric soil moisture content; VWC) were recorded on four days corresponding to the <sup>15</sup>N injection time points. These measurements were taken at four locations within each plot using a hand-held thermometer and an ML3 ThetaProbe (Delta-T Devices, Cambridge, U.K.), respectively. Bulk density was determined using intact soil cores (5 cm length, 7.2 cm diameter, 203.6 cm³ volume), collected by block (N = 8), and oven dried at 100 °C for 24 hours.

#### 2.5 DNA extraction and qPCR of the ITS region and 16S rRNA and nitrification-associated genes

The frozen, sieved soil was freeze-dried and then ground for 2 minutes using a ball mill. The DNA was extracted from 0.25 g of the milled soil using the NucleoSpin soil kit (Macherey-Nagel, Duren, Germany), with SL2 buffer with enhancer and according to the manufacturer's protocol. DNA was quantified using the Qubit 2.0 fluorometer (Invitrogen, Thermo Scientific). qPCR was used to determine the size of total bacterial and fungal communities in the soil by quantifying 16S rRNA gene and ITS, respectively. Additionally, nitrification-associated functional genes were quantified, including *amoA* (encoding ammonia monooxygenase from archaeal (AOA) and bacterial (AOB) ammonia oxidizers, and clade A (comaA) and clade B (comaB) complete ammonia oxidizers in the Nitrospira genus) and *nxrB* (encoding nitrite oxidoreductase from either Nitrospira-type (NIS) or Nitrobacter-type (NIB) nitrite oxidizing bacteria) (Table S3). Quantification was done using the C1000<sup>TM</sup> Thermal

Cycler CFX96<sup>TM</sup> Real-Time System, and CFX Connect<sup>TM</sup> Real-Time System (BioRad, CA, USA). All reactions were carried out in duplicate with a 15 μL reaction volume containing 0.1 mg/mL BSA, 1x SYBR Green Supermix (BioRad), 0.2-1.0 μM of each primer (Table S3), and 6 ng of template DNA. Standard curves were generated for each gene using serial dilutions (10<sup>2</sup>-10<sup>8</sup> copies/μL) of linearized plasmids containing the target genes. The cycling conditions, primer sequences, and concentrations for each gene are available in Table S4. The amplifications were validated by melting curve analyses and agarose gel electrophoresis. Prior to quantification, potential inhibition of PCR reactions was checked by amplifying a known amount of the pGEM-T plasmid (Promega, Madison, WI, USA) using plasmid specific M13 primers and addition of soil DNA or non-template controls for each sample. No inhibition was detected with the amount of DNA used. Gene copy numbers were adjusted for the amount and concentration of extracted DNA and normalized per gram of dry soil.

## 2.6 Data analysis

All statistical analyses were conducted in R (R Core Team, 2024) with RStudio interface (Posit team, 2024), except for the Isotope tracing model described below. R packages used included *tidyverse* (v2.0.0, Wickham et al., 2019), *rstatix* (v0.7.2, Kassambara, 2023b), *knitr* (v1.45, (Xie, 2023), *kableExtra* (v1.4.0, (Zhu, 2024), *ggpubr* (v0.6.0, Kassambara, 2023a), *sjPlot* (v2.8.15, Lüdecke, 2023). Additional R packages are described within the methods below.

#### 2.6.1 Vegetation diversity

Vegetation data from both surveys were analyzed using Correspondence Analysis (CA) to explore the relationships between time and treatment. One EcM/ErM plot was removed from the analysis because it had a vascular plant species richness of zero based on the point-framing survey in 2019. This plot had 88 % bryophyte cover, and although *Salix herbacea*, *S. polaris*, and *Empetrum nigrum* were still present there were no direct hits from the point-framing survey, indicating a presence of less than 1 % coverage.

#### 2.6.2 Isotope (15N) tracing model

Process-specific gross N transformation rates were quantified using the <sup>15</sup>N tracing model *Ntrace* (Müller et al., 2007; Rütting and Müller, 2007). We used a model setup, including three N pools (organic N, NH<sub>4</sub><sup>+</sup> and NO<sub>3</sub><sup>-</sup>) and four N transformation processes: mineralization of organic N (M<sub>Norg</sub>), immobilization of NH<sub>4</sub><sup>+</sup> and NO<sub>3</sub><sup>-</sup> (I<sub>NH4</sub> and I<sub>NO3</sub>) and NH<sub>4</sub><sup>+</sup> oxidation (O<sub>NH4</sub>, i.e. nitrification), which was sufficient to represent the observed N and <sup>15</sup>N dynamics. As we did not observe any <sup>15</sup>N enrichment of NH<sub>4</sub><sup>+</sup> following the addition of <sup>15</sup>N labelled NO<sub>3</sub><sup>-</sup>, DNRA was not considered in Ntrace. The N transformations were described by first-order kinetics, except for M<sub>Norg</sub>, which followed zero-order kinetics. The kinetic parameters of the N transformations were approximated numerically for each treatment separately with Monte Carlo sampling through a random

walk aiming to minimize a misfit function (quadratic weighted error) between the modelled and observed values. Model fit to observed NH<sub>4</sub><sup>+</sup>, NO<sub>3</sub><sup>-</sup>, and their respective <sup>15</sup>N enrichments was visually assessed (Fig. S1-S5). Model inputs were mean values and standard deviations of NH<sub>4</sub><sup>+</sup> and NO<sub>3</sub><sup>-</sup> content and their respective <sup>15</sup>N abundances. The initial <sup>15</sup>N content of the organic N pool was not measured at the plots and was instead assumed to be at natural abundance (0.366 %). Iterative approximation of the N cycle rates creates normally distributed probability density functions, for which the mean values and standard deviations were calculated (Müller et al., 2007). For pathways described by first-order kinetics, gross N rates were calculated as the product of the kinetic factor and substrate content. *Ntrace* and the optimization algorithm were set up in Matlab version R2023b and Simulink version 23.2 (The MathWorks Inc.). Rates are reported per gram of C to account for differences in organic matter content across soils and to facilitate better comparison.

The *Ntrace* provides robust estimates of gross N transformation rates but was here applied to treatment averages, hence did not allow investigation of potential block effects. To do so, we additionally quantified gross N mineralization and nitrification for each plot based on the isotope pool dilution (IPD) principle and the analytical tracing model by (Kirkham and Bartholomew, 1954) using the first two timesteps of the <sup>15</sup>N tracing experiment. All gross N transformation rates are normalized for the soil C content. To assess potential block effects on gross N mineralization and nitrification rates, we fitted generalized linear models (GLMs) with Block as a fixed effect using the *glmmTMB* package (v1.1.8, Brooks et al., 2017). Given the right-skewed distribution of the data, a zero-inflated Gamma distribution with a log link function was used for mineralization rates, while a standard Gamma distribution was applied for nitrification rates. Model significance was assessed using Type II Wald chi-square tests.

#### 2.6.3 Soil characteristics and microbial genes

To analyse the impacts of mycorrhizal status and vegetation composition on soil characteristics and microbial genes, we fitted Generalized Linear Mixed Models (GLMMs) with *glmmTMB* (v1.1.8, Brooks et al., 2017). Each model included Treatment as a fixed effect and Block as a random effect. Given that Block showed significant effects for several response variables, additional GLMs were fitted with Block as a fixed effect to explore its specific influence. These results are presented in the supplementary material for completeness, though Block was not originally intended as a primary focus of the experimental design. We validated model assumptions using the *DHARMa* package (v0.4.6, Hartig, 2022), which simulates scaled quantile residuals. Model fit was assessed through residual vs. fitted plots, QQ plots, and DHARMa's tests for uniformity and outliers to detect deviations from normality and heteroscedasticity. Pairwise comparisons between treatments were conducted with *emmeans* (v1.10.0, Lenth, 2024).

We conducted paired samples Wilcoxon signed-rank tests with the *wilcox.test* function within the *stats* package (R Core Team, 2024) to assess differences in log-transformed gene abundances between the sample groups ITS and 16S rRNA, AOA and AOB, ComaA and ComaB, as well as NIB and NIS.

We utilized the *corr.test* function within the *psych* package (v2.4.1, Revelle, 2024) to conduct correlation analyses to explore the relationships between gene abundances and environmental variables. We calculated Spearman rank-order correlation coefficients to quantify the strength and direction of these relationships. To address the issue of multiple testing and control the family-wise error rate, we applied a Holm correction. We categorized correlation coefficients based on their strength: weak  $(0 < |\mathbf{r}| < 0.4)$ , moderate  $(0.4 < |\mathbf{r}| < 0.7)$ , and strong  $(|\mathbf{r}| > 0.7)$ .

Principal Component Analysis (PCA) was employed for dimensionality reduction. The first three Principal Components (PCs) were retained, and ANOVAs were performed on them, incorporating Treatment and Block as fixed effects. The ANOVA outputs provided adjusted p-values, which were further examined using Tukey tests to identify significant differences between treatment groups and blocks.

#### 3 Results

#### 3.1 Vegetation diversity treatment effect

The treatments clearly shifted the plant community in three directions within the ordination space from its original structure in 2015 (Fig. 1). The AM/NM community and the Dominant community clustered together, whereas the EcM/ErM community and the Rare community formed their own distinct clusters after clipping treatment. The control plots in 2019 remained similar to the plant communities recorded in 2015 before the experiment was established.

Figure 1: Changes in plant communities over the course of the experiment. Mean values ( $\pm$  85 % confidence interval corresponding to an  $\alpha$ = 0.05 test; see (Payton et al., 2000, 2003) of sample scores from the correspondence analysis (CA), comparing the abundances of plant species before treatment in 2015 and four years after treatment in 2019. The eigenvalues are 0.499 for Axis 1 and 0.469 for Axis 2. Axis 1 explains 10.68 % of the total variance, and Axis 2 explains 10.04 %, together accounting for 20.72 % of the total variance. Treatments: Ctrl = control; AM/NM = plants with arbuscular mycorrhizal association or no mycorrhizal association; EcM/ErM = plants with ectomycorrhizal and ericoid mycorrhizal associations; Dominant = rare plant species removed, allowing the eight most dominant plant species to grow in the plots; and Rare = dominant species removed, keeping the eight rarest plant species.

#### 3.2 Soil characteristics

During the labelling period, VWC was significantly higher in EcM/ErM-dominated plots compared to the control (z = 2.19, p = 0.029).  $T_{soil}$  was significantly lower in Dominant plots relative to the control (z = -2.44, p = 0.015) and marginally lower in EcM/ErM plots relative to the control (z = -1.88, p = 0.06). Pairwise comparisons showed that  $T_{soil}$  in AM/NM was significantly higher than EcM/ErM (estimate = 0.02, SE = 0.006, p = 0.007) and Dominant (estimate = 0.03, SE = 0.008, p = 0.003). SOM was significantly lower in AM/NM-dominated plots compared to control (z = -2.35, p = 0.019). No other significant differences were found for the remaining soil characteristics (Table 1, Table S5). The natural abundance  $\delta^{15}$ N of SOM was measured by block and ranged from -0.08 to 2.62 (Table 1).

Table 1: Soil properties at the Tarfala study site (Sweden). Variables: soil moisture (VWC), soil temperature ( $T_{soil}$ ), laboratory gravimetric soil water content (GWC), soil organic matter (SOM), pH, C:N ratio, TN. Values represent mean  $\pm$  standard error (N = 32). VWC and  $T_{soil}$  are averaged values taken over four days of measurements, while all other properties are based on one measurement per soil sample collected from each plot. Treatments: only ecto- and ericoid mycorrhiza plant associations present (EcM/ErM), only arbuscular and non-mycorrhiza associations present (AM/NM); removal of dominant plant species (Rare); removal of rare plant species (Dominant). Significant differences from control are bolded (\* = p<0.05, \*= p<0.1) based on general linear mixed-effects models (GLMMs) (Table S5).

| Treatment | n | VWC<br>(%) | Tsoil (°C)     | GWC<br>(g/g) | SOM<br>(%) | pН      | C/N      | TN<br>(%) |
|-----------|---|------------|----------------|--------------|------------|---------|----------|-----------|
| Control   | 8 | 26.0±1.6   | $10.7 \pm 0.1$ | 55.7±1.9     | 37.2±2.5   | 4.9±0.1 | 15.3±0.5 | 1.1±0.2   |
| AM/NM     | 8 | 26.6±1.1   | 10.9±0.1       | 52.0±2.3     | 29.9±2.7*  | 4.9±0.1 | 15.6±0.6 | 0.9±0.1   |
| EcM/ErM   | 8 | 29.8±1.9*  | 10.4±0.2#      | 54.9±1.4     | 33.7±1.6   | 5.0±0.1 | 14.2±0.6 | 1.4±0.4   |
| Dominant  | 4 | 27.4±2.8   | 10.4±0.2*      | 57.6±4.4     | 38.1±5.9   | 5.0±0.0 | 16.7±0.7 | 0.9±0.1   |
| Rare      | 4 | 31.4±2.8   | 10.5±0.3       | 53.8±1.8     | 31.4±3.6   | 5.0±0.1 | 14.9±0.7 | 1.0±0.2   |

Bulk density (g cm<sup>-3</sup>) was measured by block, not by treatment: A) 0.18, B) 0.22, C) 0.21, D) 0.22, E) 1.01, F) 0.37, G) 0.41, H) 0.36. Natural abundance  $\delta^{15}$ N was measured by block, not by treatment: A) 0.95, B) 1.76, C) -0.08, D) 0.92, E) 2.62, F) 2.16

### 3.3 Gross nitrogen dynamics

Compared to the control, all treatments showed significantly higher gross N mineralization rates (Fig. 2a). EcM/ErM and Dominant showed the largest increases, at 73 % and 78 % above the Control, respectively, while AM/NM and Rare had more moderate increases of 30 % and 46 %. Gross nitrification rates were 1-2 orders of magnitude lower than gross N mineralization rates (Fig. 2b). Significant differences in gross nitrification rates were also observed, with EcM/ErM showing a 26 % increase, while Rare, AM/NM, and Dominant exhibited reductions of 32 %, 46 %, and 49 %, respectively, compared to the Control (Fig. 3b).

Average gross N mineralization and nitrification rates, calculated using the IPD approach, showed a similar pattern to those obtained through *Ntrace* (Table S6). However, the IPD based rates had much higher variability. In some instances, we even observed implausible negative rates. Block effects on gross N transformation rates were not statistically significant for mineralization ( $X^2$  (7) = 9.19, p = 0.24), but marginally significant for nitrification ( $X^2$  (7) = 13.08, p = 0.07).

Figure 2: Gross N mineralization and nitrification rates (mean and 85 % confidence interval). Rates were quantified by the *Ntrace* model with different manipulated vegetation (Control = no manipulation; AM/NM = plants with arbuscular mycorrhizal association or no mycorrhizal association; EcM/ErM = plants with ectomycorrhizal and ericoid mycorrhizal associations; Dominant = rare plant species removed allowing the eight most dominant plant species to grow in the plots; and Rare = dominant species removed keeping the eight rarest plant species). Different lowercase letters above the bars indicate significant differences based on whether the 85 % confidence intervals overlap.

#### 3.4 Abundance of bacteria and fungi

The bacterial 16S rRNA gene copy numbers were consistently higher than fungal ITS rRNA gene copy numbers across all treatments (EcM/ErM, AM/NM, Rare, Dominant), ranging from 2.07 x  $10^9$  to 2.75 x  $10^9$  and 1.05 x  $10^8$  to 1.68 x  $10^8$  copies  $g^{-1}$  dry soil respectively (V = 528, p 

Figure 3: Soil gene abundance ratios in response to plant removal treatment. Gene abundance ratios for A) fungi (ITS) vs bacteria (16S rRNA gene); B) Archaeal ammonia oxidizer (AOA) vs bacterial ammonia oxidizer (AOB); C) comammox bacteria, clade ComaA vs comammox clade ComaB; and D) nitrite-oxidizing Nitropira (NIS) vs Nitrospira (NIB). Treatments: no manipulation = Control; removal of plants with ecto and ericoid mycorrhiza associations = AM/NM; removal of plants with arbuscular mycorrhiza & no mycorrhiza associations = EcM/ErM; removal of rare plant species = Dominant; removal of dominant plant species = Rare. Symbols above the boxplots denote significant differences for each group relative to a control group as determined through Generalized Linear Mixed Models (GLMMs) (\* < 0.05) (Table S10).

#### 3.5 Nitrifier gene abundance

We observed the most notable variations in nitrification gene copy numbers between functional groups capable of the same transformation step in nitrification. Gene abundances exhibited distinct differences between functional group pairs: AOA > AOB (V = 527, p < 0.001, n = 32); NIS > NIB (V = 0, p < 0.001, n = 32); and ComaA > ComaB (V = 528, p 

Figure 4: Gene abundances representing six functional groups involved in nitrification in response to plant removal treatment. A)

Ammonia-oxidizing archaea (AOA), B) ammonia-oxidizing bacteria (AOB), C) complete ammonia oxidizers (comammox) clade A (comaA), D) complete ammonia oxidizers (comammox) clade B (comaB), E) nitrite-oxidizing Nitrobacter (NIB), and F) nitrite-oxidizing Nitrospira (NIS). Treatments: no manipulation (Control); removal of plants with ecto and ericoid mycorrhiza associations = AM/NM; removal of plants with arbuscular mycorrhiza & no mycorrhiza associations = EcM/ErM; removal of rare plant species = Dominant; removal of dominant plant species = Rare. Symbols above the boxplots denote significant differences for each group relative to a control group as determined through Generalized Linear Mixed Models (GLMMs) (\*\* < 0.01, \*\* < 0.1) (Table S12).

#### 3.6 Relationships between gene abundances, vegetation, and edaphic factors

We found no significant correlations between gene abundances and Simpson's diversity index of plants, VWC, GWC,  $T_{soil}$ , SOM, pH, TN, C/N, and BD after adjusting for multiple testing (Table S13). However, we observed a strong positive correlations between abundance of 16S rRNA genes and ITS (r = 0.73, p < 0.01), AOB (r = 0.73, p < 0.01), ComaA (r = 0.77, p < 0.01), NIB (r = 0.75, p < 0.01), AOA and NIS (r = 0.81, adj.p < 0.01), ComaA and ComaB (r = 0.72, p < 0.01), and moderate positive correlations between 16S rRNA genes and ComaB (r = 0.67, adj.p < 0.01), ITS and AOB (r = 0.69, p < 0.01), AOB and ComaA (r = 0.63, p = 0.02), ComaA and NIB (r = 0.63, adj.p = 0.02), and ComaB and NIB (r = 0.60, p = 0.04) gene abundances. We also observed moderate positive correlations between Simpson's diversity index of plants and VWC (r = 0.63, adj.p = 0.02), VWC and BD (r = 0.60, adj.p = 0.04), and BD and elevation (r = 0.60, adj.p = 0.04). There was a marginally moderate negative correlation between  $T_{soil}$  and BD (r = -0.58, adj.p = 0.08).

When considering the combined effects of vegetation diversity, soil characteristics, and the abundance of bacterial, fungal, and nitrifier genes, the first three principal components accounted for 58.2 % of the total variance, with PC1, PC2, and PC3 explaining 25.9 %, 18.5 %, and 13.7 %, respectively (Fig. S6, S7, S8). The loadings for each component indicate that no single variable drives the variance (Table S14). The strongest negative loadings on PC1 were for the abundance of ITS copies, 16S rDNA genes, and the groups ComaA, NiB, and ComaB. There were no strong positive loadings on PC1 (all were < 0.18). On PC2, the strongest positive loadings were vegetation diversity, VWC, and the abundance of AOA and NIS, while C/N ratio had the strongest negative loading. For PC3, T<sub>soil</sub> was the strongest negative loading, and elevation, VWC, GWC, and BD were the strongest positive loadings. For PC1, neither Treatment ( $F_{(4,20)} = 0.81$ , p = 0.53) nor Block ( $F_{(7,20)} = 1.28$ , p = 0.31) had a significant effect on the PC1 scores. For PC2, Treatment showed a significant effect on PC2 scores ( $F_{(4,20)} = 3.40$ , p = 0.028), while Block was not significant ( $F_{(7.20)} = 1.52$ , p = 0.22). Tukey's test indicated a significant difference between the EcM/ErM and Dominant (adj.p = 0.015), and a notable difference between Rare and Dominant (adj.p = 0.079) treatment groups. For PC3, Treatment ( $F_{(4,20)} = 3.71$ , p = 0.02) and Block ( $F_{(7,20)} = 12.01$ , p = 0.00) showed a significant effect on PC3 scores. Tukey's test indicated a significant difference between the EcM/ErM and AM/NM (p = 0.039), and between Rare and AM/NM treatment groups (p = 0.038). Tukey's test also showed significant differences between Blocks with Blocks A-D showing negative PC scores and Blocks E-H showing positive scores (Fig. S7). Elevation, which increased from Block A to H (Table S15, Fig. S9a), influenced multiple properties despite the subtle 15-meter gradient. The proportional cover of EcM-ErM plots decreased from Block A-G but deviated in Block H, where the cover resembled that of Blocks B-D. Additionally, Blocks A-D were drier than Blocks E-F, and soil temperature decreased with elevation.

#### 4 Discussion

#### 4.1 EcM/ErM communities enhance both gross mineralization and nitrification in a conservative tundra N cycle

As hypothesized, we found the highest gross N mineralization rates in the EcM/ErM treatment, but unexpectedly, the treatment with only dominant species in the plant communities also exhibited high rates. Notably, all treatments showed elevated mineralization compared to the unmanipulated control. By contrast, our hypothesis was not supported for gross nitrification. The EcM/ErM treatment was the only one showing higher nitrification rates compared to control, while all other treatments exhibited decreased rates. A previous study in a hemiboreal forest found that the presence of EcM increased gross N mineralization threefold, while gross nitrification remained largely unaffected (Holz et al., 2016). EcM-dominated ecosystems are commonly assumed to cycle N more slowly because EcM fungi promote organic N retention and decomposition of more recalcitrant substrates, whereas AM-dominated ecosystems exhibit faster N cycling due to greater reliance on inorganic N uptake and relatively fast N mineralization rates (Averill et al., 2019). However, our small-scale experimental study does not support this hypothesis, as we found significantly higher gross N cycling in the presence of EcM/ErM compared to the plots with AM/NM. This is consistent with a recent meta-analysis on rhizosphere effects on gross N mineralization (Gan et al.,

2022), demonstrating that EcM-associated plant species enhanced gross N mineralization more than AM-associated species. EcM/ErM mycorrhizal treatments circulated N faster than the other treatments, also indicated by the lower gross mineralization-to-nitrification ratio in the EcM/ErM mycorrhizal treatments (53 for EcM/ErM mycorrhizal treatments vs. 92, 81, and 134 for AM/NM and the diversity treatments). According to the mass ratio hypothesis, the plant functional traits and relative abundances of dominant species within a community are highly influential for ecosystem processes (Grime, 1998). 415 Our study partly supports the mass ratio hypothesis by demonstrating that mycorrhizal type, particularly EcM/ErM, can be regarded as a key functional plant trait influencing N cycling. However, we found that dominant species were not necessarily associated with faster or more open N cycling overall, despite high mineralization rates. The high gross mineralization-tonitrification ratio (134) in the Dominant treatment suggests a more conservative, ammonium-driven N cycle. This may reflect competitive dynamics, where dominant species more effectively acquire NH<sub>4</sub><sup>+</sup>, thereby reducing substrate availability for 420 nitrifiers. In this way, dominant species could exert a strong influence on the N cycle by both enhancing mineralization and constraining nitrification, resulting in a faster but tighter cycle that favours internal N recycling. By contrast, rare species communities exhibited lower mineralization but relatively higher nitrification (gross mineralization-to-nitrification ratio of 81), potentially indicating a more open N cycle and increased risk of N losses via leaching or gaseous pathways. These differences may arise from functional similarity and resource monopolization in dominant communities (Eisenhauer et al., 2023), versus greater functional complementarity and microbial interactions in rare communities (Niklaus et al., 2006). Thus, 425 our findings suggest that mycorrhizal status, particularly EcM/ErM associations, plays a more significant role in shaping gross N cycling dynamics than species dominance alone.

The observed increase in gross N mineralization across all manipulation treatments compared to control may be partly attributed to increased carbon input from decaying roots of plants removed by clipping. Although treatments began four years prior to our study, clipping continued during the growing season preceding it, during which a minor fraction of the removed plant species still exhibited limited regrowth. Following clipping, roots remain in the soil and decompose, potentially triggering a priming effect on the microbial community, which increases N mineralization and rhizodeposition (Bengtson et al., 2012; Dijkstra et al., 2013). Early-stage decomposition is typically rapid due to the loss of soluble carbon (Aber et al., 1990), but root decay rates decline significantly after the first year (McLaren et al., 2017). The extent of plant biomass reduction—and consequently root biomass—likely varied between treatments, with larger reductions in Rare, where all dominant plants were clipped, and smaller in Dominant. This variation may have affected labile carbon input and plant N uptake. However, the significantly higher gross N mineralization rates in the Dominant treatment, despite its similar community composition to the AM/NM, suggest that most root decaying had already subsided and had only minor effects, whereas species identity and associated functional traits drive the pattern we observe and play a-more decisive role in shaping N cycling dynamics.

Moreover, we found that gross N mineralization rates were 1-2 orders of magnitude faster than the gross nitrification rate, and the ratio of gross nitrification to  $NH_4^+$  immobilization was low. This is a strong indicator of a conservative N cycle with

minimal N losses to the environment, which is typical in N-limited ecosystems (Schimel and Bennett, 2004; Tietema and Wessel, 1992). N limitation is further supported by our  $\delta^{15}$ N data for SOM. The  $\delta^{15}$ N values of SOM depend mainly on external N sources and ecosystem N losses. In N-rich ecosystems with high denitrification, N with low  $\delta^{15}$ N is lost, resulting in higher soil  $\delta^{15}$ N values (Bai and Houlton, 2009). Conversely, in N-limited ecosystems, the primary input is via biological N fixation, which has minimal fractionation, resulting in soil  $\delta^{15}$ N values close to 0 (Amundson et al., 2003), as we observe. Few studies have investigated gross N cycling rates in situ in tundra ecosystems (Ramm et al., 2022), but our gross N mineralization rates in the control plots  $(5.0 \pm 0.3 \text{ umol g}^{-1} \text{ C d}^{-1})$  is similar to in situ rates obtained in other low Arctic and oroarctic ecosystems (Buckeridge et al., 2010; Gil et al., 2022; Paré and Bedard-Haughn, 2012). Rates on in situ gross nitrification is even more scarce for tundra ecosystems. The global average gross nitrification rate in mineral soils has been estimated to 0.56 umol g<sup>-1</sup> C d<sup>-1</sup> (Elrvs et al., 2021), whereas in permafrost mineral soils it is about half this rate, 0.27 umol g<sup>-1</sup>  $^{1}$  C d<sup>-1</sup> (Ramm et al., 2022). Our control plot nitrification rates are lower (0.13 ± 0.01 µmol g<sup>-1</sup> C d<sup>-1</sup>), and also in the lower end of what been observed in alpine grasslands, 0.16 and 0.27 umol g<sup>-1</sup> C d<sup>-1</sup> (Jin et al., 2023; Shaw and Harte, 2001). High soil C content (> 5 %) can decouple N mineralization and nitrification (Gill et al., 2023) by increasing heterotrophic N demand and intensifying competition for ammonium between heterotrophs and autotrophs (Booth et al., 2005; Keiser et al., 2016; Silva et al., 2005). Hence, our gross rates suggest that N availability in the Fennoscandian oroarctic tundra is low and low enough for the ecosystem to operate with a conservative N cycle. This leads to reduced N losses and further reinforces that N is a limiting factor controlling ecosystem productivity.

#### 4.2 Distinct soil nitrifier community within an otherwise stable microbial community

Despite the distinct roles of mycorrhizal fungi in N cycling (Castaño et al., 2023; Hobbie and Högberg, 2012; Tedersoo et al., 2020), the AM/NM and EcM/ErM plots did not differ in N-cycling gene abundances. However, altering plant composition revealed functional-differences. Notably, our Dominant community plots—although having a similar plant composition to the AM/NM plots—showed lower abundances of AOA and NIS functional groups and reduced gross nitrification rates. This may reflect stronger plant competition for NH<sub>4</sub>+ (Hayashi et al., 2016) or reduced microbial reliance on NH<sub>4</sub>+ (Hobbie and Hobbie, 2006; Schimel and Chapin, 1996). In contrast, nitrification gene abundances in Rare community plots were comparable to Control plots, despite lower gross nitrification rates. Since plant species richness was similar across Dominant and Rare treatments, our results suggest that dominant species traits—rather than richness—may drive ecosystem function, echoing findings in the ecological literature (Grime, 1998; MacGillivray et al., 1995). These traits appear to differ or be suppressed in the AM/NM community, suggesting that even a minor presence of EcM/ErM plants in an AM/NM-dominated community can shift how plant traits influence N dynamics in Arctic soils.

Although treatment effects were limited, we observed distinct communities for ammonia oxidation and nitrite oxidation. AOA was more abundant than AOB, consistent with other Arctic soils (Alves et al., 2013; Banerjee et al., 2011; Lamb et al., 2011).

Their metabolic flexibility (Alves et al., 2019), cold-tolerance (Pessi et al., 2022), and adaptation to low-N (Di et al., 2010; Erguder et al., 2009) and acidic soils (Gubry-Rangin et al., 2010; Prosser and Nicol, 2012) highlights their important role in Arctic N cycling. Among comammox clade A (ComaA) was more abundant than clade B, consistent with clade A's known adaptation to fluctuating oxygen conditions (Han et al., 2024; Palomo et al., 2018). Although comammox is underexplored in Arctic soils (Guo et al., 2024), clade B dominates nitrification in coastal Antarctica (Han et al., 2024). For nitrite oxidation, NIS was more abundant than NIB, reflecting NIS's advantage under low-nitrite conditions, where its periplasmic localization provides a competitive advantage (Nowka et al., 2015) but greater sensitivity to environmental fluctuations (Wilks and Slonczewski, 2007). Our results show a distinct nitrifier community and suggest that Arctic soils favour a more resource-efficient, yet environmentally responsive, ammonia and nitrite oxidation strategy, supporting our findings of a conservative N cycle. Moreover, we observed correlations between nitrification genes (Table S12), including a strong positive correlation between AOA and NIS, suggesting potential synergistic interactions (Jones and Hallin, 2019; Ke et al., 2013; Stempfhuber et al., 2016) within the microbial community. This reinforces the idea that N cycling in these soils is structured by microbial traits and environmental pressures rather than competitive interaction with plants and mycorrhizal fungi.

#### 4.3 Mismatch between gene abundances and in-situ activity

We found a mismatch between genetic potential for nitrification and in situ activity (gross nitrification rates) in the mycorrhizal manipulated plots. Although higher gene abundances sometimes can correlate with nitrification potential and rates (Ke et al., 2013; Laffite et al., 2020; Ribbons et al., 2016; Rocca et al., 2015), similar inconsistencies as in our study have been observed in high-Arctic soils, where the abundances of ammonia-oxidizing archaea (AOA) and ammonia-oxidizing bacteria (AOB) do not always correlate with ammonia oxidation potential (Hayashi et al., 2016). Thus, gene abundance alone does not necessarily predict nitrification rates, as environmental factors (Avrahami and Conrad, 2003; Hicks et al., 2020b; Hu et al., 2014; Li et al., 2020a; Oshiki et al., 2016; Rousk et al., 2010; Stempfhuber et al., 2016; Taylor and Mellbye, 2022; Wright and Lehtovirta-Morley, 2023), and competition (Huang et al., 2024; Jung et al., 2022; Yang et al., 2022) likely play an interacting role. Additionally, our gene targets did not encompass alternative N sources, for example N fixation (Castaño et al., 2023) or the full nitrification potential of the soil. For example, *Nitrotoga*, a cold-adapted genus of nitrite-oxidizing bacteria (NOB) (Alawi et al., 2007), competes with our targeted groups of NOB (NIB and NIS) (Alawi et al., 2009; Karkman et al., 2011; Nowka et al., 2015), but was not included in our study. Methodological choices may also explain such mismatches. DNA-based approaches reveal functional potential but cannot distinguish between living, dead, or metabolically active organisms (Burkert et al., 2019; Hansen et al., 2007; Yang et al., 2022). In contrast, RNA-based techniques provide a closer proxy for microbial activity and show stronger correlations with measured rates of key metabolic processes, including nitrification under isotopelabelled conditions (Orellana et al., 2019). Therefore, RNA-based approaches may better link functional potential with microbial process rates.

#### 4.4 Limited impact of environmental factors

Overall, neither mycorrhizal type nor plant species richness treatments had a strong influence on soil properties, nor did soil properties affect nitrification gene abundances. This result was unexpected, as above- and below-ground processes are often considered interconnected (Wardle et al., 2004). Changes in mycorrhizal type and vegetation typically influence soil properties (Netherway et al., 2021; Welker et al., 2024; Wurzburger and Brookshire, 2017), and shifts in soil conditions, management practices, or environmental conditions can affect N dynamics (Björk et al., 2007; Li et al., 2020b) and nitrification gene abundances (Zhan et al., 2023). However, vegetation is not always the primary driver of N dynamics; other environmental factors, like soil moisture, can play a more important role (Fisk et al., 1998). Recent studies suggest that below-ground communities and functions can resist changes in vegetation cover and diversity (Fanin et al., 2019; Kirchhoff et al., 2024). Consistent with this, we found no clear environmental drivers of gene abundance (Table S13). However, we observed relationships among environmental factors; vegetation diversity was positively correlated with VWC measured during the week of the labelling, while soil bulk density was positively related to elevation and VWC but negatively related to soil temperature. These relationships may be temporally dynamic, as soil moisture can strongly influence N transformation rates earlier in the growing season, with its effect diminishing later in the season (Steltzer and Bowman, 1998). Notably, our soil samples were collected during the late growing season. When analyzing vegetation diversity, soil characteristics, and gene abundances together, clear treatment differences emerged. Differences were observed between EcM/ErM and Dominant, and to a lesser extent between Dominant and Rare, driven by vegetation diversity, VWC, AOA and NIS abundances, and C:N ratio. Differences between AM/NM and EcM/ErM, and AM/NM and Rare were driven by soil temperature, elevation, VWC, GWC, and soil bulk density. Furthermore, the minimal treatment effects on water-related variables suggest that evaporation and evapotranspiration had limited influence on our results. Block effects also emerged as a key factor. There were distinct and subtle environmental gradients represented in elevation change (over a short 15-meter gradient), vegetation cover, and soil characteristics (Table S15, Fig. S9a,b). We accounted for this by including Block as a random effect, but uneven replication limited our ability to incorporate additional spatial covariates. Thus, while our design minimized disturbance, it constrained our capacity to fully separate spatial from treatment effects. Similar block effects were observed in another plant removal study involving plant-mycorrhizal associations (Kirchhoff et al., 2024), even after two years of treatment. Notably, our study spanned four years, further highlighting the persistence of these spatial influences, even within our small study area.

#### **5 Conclusions**

Our study reveals that EcM/ErM mycorrhizal associations significantly enhance N cycling in Oroarctic tundra, challenging the conventional view that EcM-dominated ecosystems cycle N more slowly. Elevated gross N mineralization rates in EcM/ErM plots suggest that these fungi are more efficient at accessing and mobilizing N from organic matter. Despite stable microbial communities, the AM/NM plots showed reduced fungal abundance, reflecting the dominance of EcM/ErM fungi in

Arctic soils. Distinct communities for ammonia and nitrite oxidation emerged, with AOA being more abundant than AOB and NIS more abundant than NIB. This supports a resource-efficient, yet environmentally responsive, N cycling strategy in these soils. However, a mismatch between gene abundances and nitrification rates suggests that environmental factors and biological competition play significant roles. Altering plant diversity revealed differences in nitrification gene abundances, with dominant plots showing lower AOA and NIS gene abundances, indicating that dominant plant species may suppress or outcompete nitrifiers. Our findings emphasize the importance of EcM/ErM in N cycling and provide a deeper understanding of ecosystem processes in tundra environments. Future research should focus on long-term experiments and monitoring to better understand how changing plant diversity and mycorrhizal associations under varying climatic conditions affect ecosystem functioning.

# Code/Data availability

All of the data are published within this paper and in the Supplement. The raw data and scripts used to make the tables and figures are available on request.

#### **Author contribution**

CRediT authorship contribution statement

AP: investigation, data curation, formal analysis, visualization, original draft preparation, review and editing. LR: conceptualization, investigation, data curation, validation, supervision, original draft preparation, review and editing. TR: conceptualization, investigation, data curation, formal analysis, visualization, validation, supervision, original draft preparation, review and editing. SB: investigation, validation, review and editing. SH: conceptualization, methodology, funding acquisition, investigation, validation, review and editing. JJ: investigation, validation, review and editing. CFS: resources, investigation, review and editing. MPB: supervision, review and editing. PB: review and editing. GR: conceptualization, resources, review and editing. RGB: conceptualization, methodology, project administration, funding acquisition, investigation, formal analysis, visualization, original draft preparation, review and editing.

#### 565 Competing interests

The authors declare that they have no conflict of interest.

#### Acknowledgements

The authors would like to show gratitude to the staff at Tarfala Research Station for their help and services. We would also like to send a big thanks to the field assistants Haldor Lorimer-Olsson, Stina Johlander, Saskia Bergmann, Josefina Pehrson, Lasse Keetz, and Erik Bergström. The paper was funded by the Swedish Research Council (Vetenskapsrådet 621-2014-5315 to RGB) and was a contribution to the Strategic Research Area "Biodiversity and Ecosystem Services in a Changing Climate" (BECC) funded by the Swedish government. This study has been made possible by data and support provided by the Swedish Infrastructure for Ecosystem Science (SITES).

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
