# Peer review of "The role of mycorrhizal type and plant dominance in regulating nitrogen cycling in Oroarctic soils"

_EGUsphere, 2025_

## Author Response (AR1)

**Reply to Associate Editor**

Public justification (visible to the public if the article is accepted and published): Dear Dr. Robert G. Björk,

I have read your point-by-point rebuttal letter in response to the comments by two reviewers and I am happy for you to incorporate them in a revised version of your manuscript. In your revision please clarify the association between mycorrhizal types and dominant vs. rare species in the different treatments and consider/discuss the reduction in plant biomass associated with each treatment. Discuss also the implications of spatial heterogeneity and the limitations of the study design.

| Thank you.     |
|----------------|
| Sincerely,     |
| Erika Buscardo |

Reply: We thank you for the opportunity to revise our manuscript and for the constructive feedback. In the revised version, we have incorporated most of the reviewers' suggestions and clarified the rationale behind our treatments. Specifically, we now discuss the association between mycorrhizal types and dominant versus rare species, the reduction in plant biomass across treatments, and the implications of spatial heterogeneity. We believe that our interpretation of the results and the limitations of the study are now more nuanced and clearly presented.

**Reply to Anonymous Referee #1**

**General comments**

This study deals with effects of plant composition on soil nitrogen turnover in shrub tundra, a relevant topic considering the ongoing changes in vegetation composition and nutrient turnover associated with climate change in the arctic. It is a carefully designed and conducted study with clear objectives and finalized appearance. The motivation for the study as well as interpretation of results are well-argumented and referenced, and all the content is relevant, making it a nice and compact read. At places, some additional explanation would be good, as specified below.

Reply: We thank reviewer 1 for their encouraging assessment of our manuscript. We are pleased that the study's relevance, design, and clarity came through, and we appreciate your positive remarks on the motivation, interpretation, and overall structure. We have considered your suggestions for additional clarification and have addressed them point

by point in our responses below. We hope the revisions improve the manuscript and meet your expectations.

**Major/intermediate comments**

1. Some additional info about plant removal treatments would be needed:

It would be good to clarify the association between the ErC/ErM and AN/MN vs. dominant and rare treatments. Where dominant species mainly ErC/ErM or AN/MN plants, or en even mixture of the two types? How about rare? Knowing this might be relevant for interpreting the results, so it would be good to comment this in the text and perhaps add to Supplementary Table S1.

Reply: We thank the reviewer for this insightful comment. The Dominant and Rare treatments were designed to include an even mixture of mycorrhizal types (ErM/EcM and AM/NM) to allow us to disentangle the effects of species richness from those of mycorrhizal association. We agree that clarifying this in the manuscript will help readers better interpret the results.

Data show that both the Rare and Dominant treatments included a relatively balanced removal of species across mycorrhizal types. For example, in the Rare treatment, species removals included 6 ErM, 5 EcM, 7 AM, and 6 NM species. Similarly, the Dominant treatment included the removal of 7 ErM, 5 EcM, 7 AM, and 10 NM species. While the Dominant treatment removed slightly more NM species than the Rare treatment, both treatments maintained a mix of mycorrhizal types.

Regarding the species remaining after treatment, Dominant plots contained an average of 12 species, with 8 AM/NM species (31% cover) and 5 EcM/ErM species (43% cover). Rare plots contained an average of 10 species, with 8 AM/NM species (7% cover) and only 1 EcM/ErM species (<1% cover). These differences in cover reflect the natural dominance of certain species but do not indicate a systematic bias in the design of the treatments.

We have updated Supplementary Table S1 to include this information and revised the sentence "Rare and dominant plant species were determined by reducing species richness to approximately 50 % of the community species pool while maintaining a mixed mycorrhizal plant community." formerly on Lines 119-120 for clarification.

Lines 119-120 now appear on Lines 132-136 in the tracked changes document and read:

"The Dominant and Rare species removal treatments were designed to include a relatively even mixture of species representing different mycorrhizal types (EcM, ErM, AM, and NM). This design allowed us to separate the effects of species richness from

those of mycorrhizal association. While the exact number of species removed varied slightly between treatments, both included a balanced representation of mycorrhizal types."

Further, it would be worth commenting, how large plant biomass reduction was associated with each treatment. I would assume that this was highly variable between treatments, particularly between Rare (larger reduction) and Dominant (higher reduction). This reduction of plant biomass would be associated with reduction of labile C input and plant N uptake, which both may be important for soil N cycling. The focus of this study was in quality (= changes in plant composition), many aspects of C and N cycling are driven by simple quantity and this should be acknowledged.

Reply: We appreciate the reviewer's point regarding the importance of biomass quantity in driving soil C and N cycling. While we did not quantify aboveground biomass reductions directly, our experimental design was conceptually grounded in the mass-ratio hypothesis (Grime 1998), which posits that dominant species influence ecosystem processes primarily through their biomass-scaled traits, whereas rare species exert influence more through functional identity than quantity.

Thus, the contrast between dominant and rare species in our study can be interpreted as a test of these two mechanisms:

- For dominant species, biomass likely represents a key mechanism driving changes in soil processes.
- For rare species, the effects are more likely tied to species identity and trait novelty, rather than biomass per se.

We have tried to clarify this conceptual framing in the introduction (Line 57-63) to improve the linkage to the discussion where we previously only referred to the mass ratio hypothesis. In addition, we acknowledge in the discussion (lines 441-456) that differences in biomass reduction occur, but that any potential effect of this most likely has subsided over the four years of clipping prior to our experiment.

**Lines 57-63 now reads as follows:**

"According to the mass ratio hypothesis (Grime, 1998), ecosystem processes such as primary production, nutrient cycling, and soil microbial composition, are primarily driven by dominant plant species, whose high biomass and resource use exert a disproportionate influence—while the contributions of rare species are considered minimal (Grime, 1998; Tedersoo et al., 2020). This disproportionate influence also extends to nitrogen dynamics, where dominant species, through their biomass-scaled

traits, can affect soil N availability by regulating N mineralization and nitrification (Clemmensen et al., 2021; Kielland, 1995; Liu et al., 2018; Michelsen et al., 1996; Ramm et al., 2022; Rozmoš et al., 2022; Tunlid et al., 2022)."

**Lines 441-456 now reads as follows:**

"The observed increase in gross N mineralization across all manipulation treatments compared to the control may be partly attributed to increased carbon input from decaying roots of plants removed by clipping. Although treatments began four years prior to our study, clipping continued during the growing season immediately preceding it, during which a minor fraction of the removed plant species still exhibited limited regrowth. Following clipping, roots remain in the soil and decompose, potentially triggering a priming effect on the microbial community that enhances N mineralization and rhizodeposition (Bengtson et al., 2012; Dijkstra et al., 2013). Early-stage decomposition is typically rapid due to the loss of soluble carbon (Aber et al., 1990), but root decay rates decline significantly after the first year (McLaren et al., 2017). The extent of plant biomass reduction —and consequently root biomass— likely varied between treatments, with larger reductions in Rare, where all dominant plants were clipped, and smaller in Dominant. This variation may have affected labile carbon input and plant nitrogen uptake. However, the significantly higher gross N mineralization rates in the Dominant treatment, despite its similar community composition to AM/NM, suggest that most root decaying had already subsided and had only minor effects, whereas species identity and associated functional traits drive the pattern we observe and play a more decisive role in shaping nitrogen cycling dynamics."

**2. A few points could be clarified regarding the 15N method and results:**

The study design would have allowed determination of DNRA rate but was not determined (line 210->). Was this decided prior to running the model because it was assumed that it is unimportant, or was it found irrelevant based on the modelling results?

Reply: Indeed, the tracing model would allow for quantifying DNRA rates. The process was excluded prior to running the model, as the measured 15N data showed no enrichment in the ammonium pool following 15N labeling of the nitrate pool.

**We have added a short statement on lines 232-233:**

"...), which was sufficient to represent the observed N and  $^{15}$ N dynamics. As we did not observe any  $^{15}$ N enrichment of NH4+ following the addition of  $^{15}$ N labeled NO3-, DNRA was not considered in *Ntrace*."

Concerning 15N label addition, the level is clearly explained on lines 145-146 but since the native mineral N pool sizes are not reported, it is not clear how large addition this was compared to the native pool.

Reply: Due to the remoteness of the research site, we could not measure the native mineral N before the  $^{15}$ N labelling experiment. As explained, we therefore based the amount  $^{15}$ N added on mineral N data from a similar site, aiming at adding 10% of the (expected) native N pool. For nitrate, this could not be achieved, as this would result in too low NO3 content for  $^{15}$ N analysis. Based on the measured initial  $^{15}$ N after labelling, we estimate that we finally added about 50 % of the native NO3 pool and < 10% of the native NH4 pool.

We have added a short sentence on lines 163-165 (previously 145-146):

"...), aiming for a  $^{15}$ N enrichment of 10%. For NO $_3$  a larger amount was added, approximately 50% of the native pool, which was required for  $^{15}$ N analysis."

Also, it would be good to show some plots with the data used to run the model. In systems with closed N cycling the rate of NH4+ and NO3- can be small compared to immobilization or initial stabilization of isotopic pools, causing some challenges for 15N labelling experiments. Based on the negative gross min rates observed, mentioned on line 301, this might have been the case. What was the approach when negative rates were observed? Were they excluded from the data set, or included? Would this issue have bene solved by using a latter time period instead of first two sampling points?

Reply: The few negative gross rates observed are rather due to the heterogeneity in the soil. Compared to laboratory-based studies, the in-situ approach with undisturbed soil cannot guarantee complete homogeneity between the time points. Particularly in systems with small gross rates, small differences in e.g. mineral N content between the soil cores can lead to negative rates. This is independent of the time step and would probably be more common using the later time points, as the differences in 15N enrichment would become smaller, making it even more susceptible to natural heterogeneity. Note also that the negative rates are only for the IPD approach but cannot occur in the Ntrace approach.

We have included supplementary figures S1-S5 to show model fit and have added a reference to them on lines 236-237, which reads as follow:

"Model fit to observed  $NH_4^+$ ,  $NO_3^-$ , and their respective 15N enrichments was visually assessed (Fig. S1-S5)."

Minor comments

Line 21: It is important to know how many years after start of the manipulation the experiment was conducted. The initial treatment effects may be very different from long-term effects. Now, this information is clearly stated only on line 411, but I recommend adding this here and, in the methods, line 137. Also, the timing of soil sampling could be specifically stated, now I assume it was at the same time as the labeling experiment but cannot be sure.

Reply: Yes, and we have clarified this in the suggested places.

**Lines 20-21 now read as follows:**

"... ecosystem. Four years after a plant removal treatment, we measured these rates using *in situ* 15N labelling and quantified a selection of nitrification..."

Line 154 (previously 137) now reads as follows:

"Four years after plant removal, gross..."

Line 179 now reads as follows:

"At the same time as the 15N labelling experiment, ..."

line 22-23: Please give the treatment names in parenthesis also for 2 and 3 (similarly as for other treatments) to avoid any confusion with the treatment names.

Reply: We have revised this and hope this is clearer now.

**Lines 21-25 now reads as follow:**

"Treatment plots included (1) unmanipulated (Control); or the removal of: (2) ectomycorrhizal (EcM) and ericoid mycorrhizal (ErM) plants, letting arbuscular mycorrhizal (AM) and non-mycorrhizal (NM) plants dominate (AM/NM); (3) AM and NM plants, letting EcM and ErM plants dominate (EcM/ErM); (4) low-abundance species, leaving the most abundant species (Dominant); and (5) high-abundance species, leaving only the low-abundance species (Rare)."

line 147: How thick was the organic layer? Was this top 6 cm fully in the organic layer or did it include the upper part of the underlying mineral soil horizon? Below in the next section, it is stated that soil characteristics apply only to organic horizon, but it is not completely clear if the whole 0-6 cm was organic.

Reply: The top 6 cm of soil represented the organic layer, and only the organic layer was used in this study. We have clarified this on both Line 164 and Lines 178–179.

Line 166 (previously 147) now reads as follows:

"..., inserted to a depth of 6 cm within the organic soil layer, at each of..."

**Lines 179-180 now read as follows:**

"At the same time of the 15N labelling experiment, we also collected samples from the top 6 cm of the organic soil layer to assess abiotic and biotic soil characteristics, matching the depth used for labelling."

line 293: In addition to the enhanced mineralization caused by decaying roots, could this higher mineralization in all plant removal treatments also result from secondary metabolites from plants that would inhibit soil microbial community mediating N-cycling (e.g., Moreau et al. 2019, DOI: 10.1111/1365-2435.13303, and the references therein)? This has been reported sometimes, and I recommend adding some discussion about this possibility to the discussion section.

Reply: We appreciate the reviewer's thoughtful suggestion and the reference to Moreau et al. (2019). The potential role of plant-derived secondary metabolites in inhibiting microbial communities involved in nitrogen cycling is indeed an interesting possibility. While we did not quantify secondary metabolites in this study, we briefly touch on this topic in our discussion (lines 446–461). Although this section could be expanded, we feel that, given the already extensive scope of the discussion and the lack of direct measurements, further elaboration would remain speculative and would not substantially alter the interpretation of our findings.

line 360->: I strongly recommend adding PCA biplots (PC1+PC2, PC1+PC3) to the supplementary materials to accompany the other plots related to PCA. That would greatly help getting an overview about treatment and block differences and understanding the text and the other graphs. Why the variable loadings are given for PC2 and PC3, but not for PC1 explaining more of the variability? The meaning of the sentence "...some variables show stronger contributions to certain components." is not clear, this sounds too obvious but maybe I am misunderstanding what is meant here.

Reply: The PCA biplots have been added as figure S8 in the supplementary information, and PC1 variable loadings have been included on lines 395-398.

**Lines 391-394 now read as follows:**

"The strongest negative loadings on PC1 were for the abundance of ITS copies,16S rDNA genes, and the groups ComaA, NiB, and ComaB. There were no strong positive loadings on PC1 (all were  $\leq$  0.18). On PC2, the strongest positive loadings were vegetation diversity, VWC, and the abundance of AOA and NIS, while C/N ratio had the

strongest negative loading. For PC3, Tsoil was the strongest negative loading, and elevation, VWC, GWC, and BD were the strongest positive loadings."

Yes, the sentence "...some variables show stronger contributions to certain components." is obvious, and we have now removed it from lines 396 (previously 387-388.)

line 405: Could the lower mineralization-nitrification ratio in the Rare treatment as compared to Dominant be a consequence of lower plant biomass -> lower N uptake -> better availability for nitrifiers from mineral N? Please see above in major comment 1.

Reply: See our reply to major comment 1.

line 425: Please give a reference to where these values are reported. They are now given under the Table 1 but not very easy to find. Since these results are referred to in the discussion section would be good to report them more clearly in the results section.

Reply: Yes, this was a bad decision made by us. We have now incorporated it in the main text under section 3.2.

On Lines 305-306, we added:

"The natural abundance  $\delta^{15}N$  of SOM was measured by block and ranged from –0.08 to 2.62 (Table 1)."

section 3.4. Small changes in community composition suggests the microbial community composition was robust for these plant removals, but activity might have changed, as suggested by gross rate determination. Calls for RNA study.

Reply: We interpret the reviewer's comment as a suggestion to highlight the need for future RNA-based studies, and we have incorporated this point into our discussion in Section 4.3 on lines 534-539 and reads:

"Methodological choices may also explain such mismatches. DNA-based approaches reveal functional potential but cannot distinguish between living, dead, or metabolically active organisms (Burkert et al., 2019; Hansen et al., 2007; Yang et al., 2022a). In contrast, RNA-based techniques provide a closer proxy for microbial activity and show stronger correlations with measured rates of key metabolic processes, including nitrification under isotope-labelled conditions (Orellana et al., 2019). Therefore, RNA-based approaches may better link functional potential with microbial process rates."

**Reply to Anonymous Referee #2**

This study experimentally manipulated Arctic plant communities to test the effects dominant mycorrhizal associations as well as dominant vs rare plant communities. The writing is generally excellent, background well organized, and hypotheses clearly define. I appreciate that some of the results were explained as surprising in context of the hypotheses.

Reply: We thank reviewer 2 for the constructive feedback. We are glad to hear that the experimental approach, organization of the background, and clarity of the hypotheses were well received. We also appreciate your recognition of our efforts to interpret unexpected findings within the context of our initial expectations. Your comments were encouraging and helpful, and we have addressed the specific points raised in your comments below.

**Intro:**

One main point I think could use addressing in the logic and setup of the study is that it's a bit hard to understand the specific treatments. Specifically, why EcM/ErM were lumped as opposed to AM/NM, and how these two treatments are more or less similar to the dominant and rare treatments. I would imagine that the shrubs tend to collocate (EcM and ErM) and then the grasses. Was the natural tendency for EcM and ErM to be found next to one a reason why there was no AM/EcM or AM/ErM treatment?

Reply: One of the motivations behind our groupings was to reflect the large-scale patterns in carbon and nitrogen dynamics identified by Averill et al. (2014) and Read & Moreno (2003), as well as the small-scale patterns described by Giesler et al. (1998) and Björk et al. (2007). EcM and ErM were grouped based on their shared functional traits—particularly their saprotrophic capabilities and ability to access organic nutrient pools in nutrient-poor soils. In contrast, AM and NM plants were grouped due to their association with faster nutrient cycling and preference for more mineral-rich soils. Tundra communities offer a unique advantage in this context, as they naturally encompass all major mycorrhizal associations within a single plant community, making them ideal testbeds for experimental manipulation. By altering these associations, we aimed to disentangle the ecological processes underlying both large- and small-scale patterns.

To ensure that observed effects are attributable to changes in mycorrhizal associations rather than shifts in species richness, as argued in the mass ratio hypothesis, we also implemented treatments that varied in species dominance and rarity. These treatments included a mixture of mycorrhizal types, allowing us to control for species composition

while simultaneously addressing a broader ecological question: the relative importance of dominant versus rare species in driving ecosystem processes.

We have now incorporated this into the objectives paragraph of the introduction (lines 96-101). To keep the paragraph concise, this also required additional revisions, which are highlighted in red.

**On Lines 96-101, we added:**

"Our mycorrhizal groupings reflect broad nutrient cycling patterns (Averill et al., 2014; Read & Moreno, 2003) and finer-scale dynamics (Giesler et al., 1998; Björk et al., 2007). EcM and ErM were grouped based on shared traits such as saprotrophic capacity and organic nutrient acquisition, while AM and non-mycorrhizal (NM) associated plants were linked by their association with faster nutrient turnover. These mycorrhizal types also naturally co-occur in tundra vegetation. To separate effects of mycorrhizal function from species richness as argued in the mass ratio hypothesis, we also varied species dominance and rarity, enabling us to test how functional traits and community structure influence ecosystem processes."

**Methods:**

I understand that to sample in each of the plots for a Before After Control Impact study would have created additional disturbance and could have affected belowground N cycling. However, I think that some kind of spatial accounting for the blocks is necessary given the finding that blocks had an effect on multiple variables. The supplemental plots are great. Did you consider using fixed effects in your main model that may have differed between block, such as landscape position that can affect longterm patterns in soil temperature and moisture? The topographic wetness index is one I would suggest that would take into account relative differences in elevation. I see in the supplement plot S3 that there is a difference in elevation (albeit a small one) from the highest to lowest plot (elevation y axis could use units), but this corresponds to differences in temperature, and to some degree soil moisture. From the PC scores in Fig S2, it looks like PC3 may follow an elevational/temperature trend.

Reply: We agree that spatial heterogeneity, even within relatively small field sites, can influence environmental variables such as temperature and soil moisture. We did consider including additional fixed effects that varied by Block, such as elevation or other topographic features like the TWI. However, our experimental design included uneven replication across treatments, which limited our statistical power and increased the risk of overfitting if too many fixed effects were included in the models. To ensure model stability and interpretability given our sample size, we prioritized parsimony in our fixed effect structure. We addressed the influence of spatial variation by including

Block as a random effect in our main GLMMs. Because block showed significant effects for several response variables, we also fitted supplementary models with Block as a fixed effect to further explore its influence.

All plots were situated within a relatively small area (~2400 m²), with plots within blocks separated by only a few meters and blocks by tens of meters. While small differences in elevation corresponded to differences in soil temperature and, to some extent, VWC, we interpret these as natural small scale spatial variability rather than systematic topographic or landscape-scale effects. Because we measured VWC directly for each plot, we felt that incorporating a landscape-scale proxy such as the TWI, whose predictive power has been shown to be modest and highly dependent on DEM resolution (Riihimäki et al. 2021), would not substantially improve our model or interpretation.

We do appreciate the suggestion and acknowledge that TWI or similar indices may be more useful in broader-scale studies. We have expanded the discussion on the block effect (lines 560-566), which now reads as follows:

"Block effects also emerged as a key factor. There were distinct and subtle environmental gradients represented in elevation change (over a short 15-meter gradient), vegetation cover, and soil characteristics (Table S14, Fig. S3a,b). We accounted for this by including Block as a random effect, but uneven replication limited our ability to incorporate additional spatial covariates. Thus, while our design minimized disturbance, it constrained our capacity to fully separate spatial from treatment effects. Similar block effects were observed in another plant removal study involving plantmycorrhizal associations (Kirchhoff et al., 2024), even after two years of treatment. Notably, our study spanned four years, further highlighting the persistence of these spatial influences, even within our small study area."

The experimental design seems well thought out and thorough. One consideration is the effect of clipping more or less vegetation in the different treatments and how this would affect water surface evaporation, evapotranspiration, and ultimately the amount of moisture in the soil that could affect N cycling. There seem to be differences in field moisture, and I wonder if the authors could briefly speak to how this disturbance could have affected processes belowground. Could even be 1-2 lines in the discussion.

Reply: To minimize the influence of evaporation and evapotranspiration on our treatments, we enclosed each plot with a 1 µm mesh cloth and positioned them on a gentle slope. This design facilitates lateral water movement across plots, helping to buffer against localized water loss. As a result, we observed minimal treatment effects on our water-related variables, suggesting that evaporation and evapotranspiration are

unlikely to have significantly influenced our results. We have added a short sentence on lines 559-560:

"Furthermore, the minimal treatment effects on water-related variables suggest that evaporation and evapotranspiration had limited influence on our results."

**Small suggestions**

The background information is well written, and often there are many citations for sentences containing multiple clauses. I would suggest moving citations so that ones for soil C content or nutrient cycling come right after the respective clause (lines 40-45 for example) to match citations with the info. It may make readability a bit harder, so up to your discretion.

Reply: Thank you for this thoughtful suggestion. We agree that aligning citations more closely with specific clauses can improve clarity in some cases. However, we also considered the potential trade-off with readability and flow. After reviewing the relevant section, we decided to retain the current citation placement.

Line 68 – Starting with AM fungi being less common in Arctic systems does not sound compelling to include as a dominant type in this experiment. You could reword to start with EcM and ErM tending to be dominant in Arctic systems, but AM still prevalent and likely still affecting N cycling etc despite the lower abundance.

Reply: Thank you for the suggestion. We have reworded the beginning of this paragraph on lines 72-74 (previously 68), which now reads as follows:

Ectomycorrhizal (EcM) and ericoid mycorrhizal (ErM) fungi tend to dominate Arctic ecosystems (Michelsen et al., 1998; Soudzilovskaia et al., 2017; Steidinger et al., 2019), whereas arbuscular mycorrhizal (AM) fungi are considered less common due to low cold tolerance but still prevalent (Kilpeläinen et al., 2016; Kytöviita, 2005; Ruotsalainen and Kytöviita, 2004; Wang et al., 2002).

Line 70. "Different mycorrhizal types" could be instead "These three mycorrhizal types" as you introduce them already and don't include orchid mycorrhizae.

Reply: Agree, we have incorporated this suggestion (now line 75).

Line 86. I would restructure the topic sentence a bit for succinctness, with something like: "We aimed to determine the relative effects of functional (mycorrhizal) and structural (rare, dominant) diversity on soil N cycling."

Reply: Thank you for the suggestion, which we have incorporated on lines 90-91.

Line 90. Would remove "thus." I'm a fan of thus and therefore, but doesn't seem necessary here. And should the hypotheses be in past tense?

Reply: Agree, that "thus" can be removed, which we have now done (now line 101). However, we chose to present our hypotheses in the present tense in the Introduction to reflect their role in framing the study rationale, in line with guidance from Josh Schimel's book Writing Science: How to Write Papers That Get Cited and Proposals That Get Funded.

I like how the methods has a data analysis section for each component. In the isotope tracing section, I am curious why the rates are relativized by per g C, and not per g N (assuming this is bulk). I don't have experience with these models, and C and N tend to increase together, but there can be interesting differences in CN. Forgive my ignorance if this is common practice.

Reply: In the literature, gross N transformation rates have both been presented on a per g C and per g N basis, besides the most common per soil dry weight. From our point of view, the per g C basis is more appropriate, as we assume that in most soil (heterotrophic) microorganisms are limited by C, and not N. Hence, the rates expressed per g C provide information on if rates differ beyond those explained by the limiting resource for microbial activity, and indirectly for nitrification.

Line 238. How did you validate models exactly? Visually or with additional fit statistics?

Reply: We validated model assumptions using the DHARMa package (line 238), which simulates scaled quantile residuals to assess model fit. We examined residual vs. fitted plots and QQ plots and used DHARMa's tests for uniformity and outliers to identify potential violations of non-normality and heteroscedasticity. We have revised the methods to clarify this approach.

Lines 261-263 (previously 238) will now read as follows:

We validated model assumptions using the DHARMa package (v0.4.6, (Hartig, 2022)), which simulates scaled quantile residuals. Model fit was assessed through residual vs.

fitted plots, QQ plots, and DHARMa's tests for uniformity and outliers to detect deviations from normality and heteroscedasticity.

Line 380. The explanation of the findings is compared well to the literature. However, for clarity, I would suggest the topic sentence be more like the last sentence of the paragraph so the reader knows where the paragraph is going. I found it to be a bit winding and unclear what the main take away was.

Reply: Our topic sentence reads "As hypothesized, we found the highest gross N mineralization rates in the EcM/ErM treatment, but unexpectedly, the treatment with only dominant species in the plant communities also exhibited high rates.", which we believe shows what the paragraph is about. The last sentence "Thus, our findings suggest that mycorrhizal status, particularly EcM/ErM associations, plays a more significant role in shaping gross N cycling dynamics than species dominance alone." concludes our findings.

Section 4.2. The focus on background info about mycorrhizal effects that your study does not support takes away from the findings you do have on how the dominant community affects functional groups. Also, towards the end near line 455, there's a cool line about how AM/NM had similar composition to dominant plots. I feel like this is worth highlighting towards the beginning of this paragraph, especially since you found differences in the dominant treatment.

Reply: Thank you for the suggestion, we have rephrased the paragraph according to the suggestions.

**First paragraph in section 4.2 now reads (lines 479-491):**

Despite the distinct roles of mycorrhizal fungi in N cycling (Castaño et al., 2023; Hobbie and Högberg, 2012; Tedersoo et al., 2020), the AM/NM and EcM/ErM plots did not differ in N-cycling gene abundances. However, altering plant composition revealed functional differences. Notably, our Dominant community plots—although having a similar plant composition to the AM/NM plots—showed lower abundances of AOA and NIS functional groups and reduced gross nitrification rates. This may reflect stronger plant competition for  $NH_4^+$  (Hayashi et al., 2016) or reduced microbial reliance on  $NH_4^+$  (Hobbie & Hobbie, 2006; Schimel & Chapin, 1996). In contrast, nitrification gene abundances in Rare community plots were comparable to Control plots, despite lower gross nitrification rates. Since plant species richness was similar across Dominant and Rare treatments, our results suggest that dominant species traits—rather than richness—may drive ecosystem function, echoing findings in the ecological literature (Grime, 1998; MacGillivray et al., 1995). These traits appear to differ or be suppressed in the AM/NM

community, suggesting that even a minor presence of EcM/ErM plants in an AM/NM-dominated community can shift how plant traits influence N dynamics in Arctic soils.

Line 460-480 This paragraph on the nitrifier community and N-cycling gene abundances seems a bit long for having no treatment effects. Does the last sentence about microbial traits and environmental pressures contradict your other findings and the next paragraph 4.3? Why do these treatments in the first place if the different plant community combinations have no effect?

Reply: We agree that the section is currently long, given the lack of strong treatment effects, and we have revised it for conciseness (reduced by about 100 words), and it reads as follows:

**Lines 493-520 (previously 460-480):**

"Although treatment effects were limited, we observed distinct communities for ammonia oxidation and nitrite oxidation. AOA were more abundant than AOB, consistent with other Arctic soils (Alves et al., 2013; Banerjee et al., 2011; Lamb et al., 2011). Their metabolic flexibility (Alves et al., 2019; Verhamme et al., 2011), cold-tolerance (Pessi et al., 2022), and adaptation to low-N (Di et al., 2010; Erguder et al., 2009) and acidic soils (Gubry-Rangin et al., 2010; Prosser and Nicol, 2012) highlight their important role in Arctic N cycling. Among comammox, clade A (ComaA) was more abundant than clade B, consistent with clade A's known adaptation to fluctuating oxygen conditions (Han et al., 2024; Palomo et al., 2018). Although comammox is underexplored in Arctic soils (Guo et al., 2024), clade B dominates nitrification in coastal Antarctica (Han et al., 2024). For nitrite oxidation, NIS was more abundant than NIB, reflecting NIS's advantage under lownitrite conditions, where its periplasmic localization provides a competitive advantage (Nowka et al., 2015) but greater sensitivity to environmental fluctuations (Wilks and Slonczewski, 2007). Our results show a distinct nitrifier community and suggest that Arctic soils favour a more resource-efficient, yet environmentally responsive, ammonia and nitrite oxidation strategy, supporting our findings of a conservative N cycle. Moreover, we observed correlations between nitrification genes (Table S12), including a strong positive correlation between AOA and NIS, suggesting potential synergistic interactions (Jones and Hallin, 2019; Ke et al., 2013; Stempfhuber et al., 2016) within the microbial community. This reinforces the idea that N cycling in these soils is structured by microbial traits and environmental pressures rather than competitive interaction with plants and mycorrhizal fungi."

However, because plant and mycorrhizal communities can influence soil microbial composition and nutrient cycling, we believe the treatments are valid and we hope that

the revisions of the introduction support this better now. The limited treatment effects on the functional guilds of nitrifiers do not rule out plant influences but indicate that abiotic constraints, such as temperature and moisture, may override biotic drivers in this context. By examining the abundance of different groups of nitrifier communities, we could conclude that guilds with resource-efficient ammonia and nitrite oxidation strategies are favoured irrespective of treatment, which gives further support to our findings of a conservative N cycle in these Arctic soils. This is not contradicting the findings discussed in 4.3, as the genetic potentials reflect more long-term capacity and the ecology of the present nitrifier guilds, while the measured rates show the activity of the time of sampling.